# Differentiable Architecture Search: a One-Shot Method?

**Jovita Lukasik**[*1] **Jonas Geiping**[*2] **Michael Moeller**[†1] **Margret Keuper**[†1,3]

[1]University of Siegen
[2]University of Maryland
[3]Max Planck Institute for Informatics, Saarland Informatics Campus

**Abstract** Differentiable architecture search (DAS) is a widely researched tool for the design of novel architectures. The main benefit of DAS is the effectiveness achieved through the weight-sharing one-shot paradigm, which allows efficient architecture search. In this work, we investigate DAS in a systematic case study of inverse problems, which allows us to analyze these potential benefits in a controlled manner. We demonstrate that the success of DAS can be extended from image classification to signal reconstruction, in principle. However, our experiments also expose three fundamental difficulties in the evaluation of DAS-based methods in inverse problems: First, the results show a large variance in all test cases. Second, the final performance is strongly dependent on the hyperparameters of the optimizer. And third, the performance of the weight-sharing architecture used during training does not reflect the final performance of the found architecture well. While the results on image reconstruction confirm the potential of the DAS paradigm, they challenge the common understanding of DAS as a one-shot method.

## 1 Introduction

Recent progress in computer vision and related fields has illustrated the importance of suitable neural architecture designs and training schemes (He et al., 2016). Ever deeper and more complex networks show promise, and manual network design is less and less able to explore the desired search spaces. Neural architecture search (NAS) (Elsken et al., 2019; White et al., 2023) is the task of optimizing the architecture of a neural network automatically without resorting to human selection, scaling to larger search spaces and proposing novel well-performing architectures. NAS has been successfully addressed using black-box optimization approaches such as reinforcement learning (Zoph and Le, 2017; Zoph et al., 2018) or Bayesian optimization (Kandasamy et al., 2018; White et al., 2021; Ru et al., 2021; Lukasik et al., 2021). However, these approaches are computationally expensive as they require the training of many candidate networks to cover the search space. In contrast, differentiable architecture search (Liu et al., 2019) proposes a continuous relaxation of the search problem, i.e., all candidate architectures within a given search space of operations and their connectivity are jointly optimized using shared network parameters while the network also learns to weigh these operations. The final architecture can then simply be determined by selecting the highest weighted operations. This is appealing as practically good architectures are proposed within a single optimization run. However, previous works such as Zela et al. (2020) also indicate that the proposed results are often sub-optimal, especially when the search space is not well chosen. We make a clear distinction here between *DARTS* (Liu et al., 2019), which includes the proposed search space for image classification models and the differentiable architecture search, *DAS*, itself.

In this paper, we apply DAS to inverse problems with the main focus on the analysis of DAS w.r.t. the impact of domain shifts, training hyperparameter choice and network initialization.

---

[*] Equal contribution.

[†] Equal contribution.

JL, MM and MK acknowledge support by the DFG research unit 5336 - Learning to Sense.

Since signal recovery has not received nearly as much attention in the NAS literature as image classification, it allows to study a naive choice of parameters and settings without bias to known results and best practices. In the signal recovery setting, sequential architectures (Zhang et al., 2017) yield competitive results when learning to solve inverse problems, such that we can analyze the impact of the complexity of the search space more easily. Specifically, we compare the stability and sensitivity to hyperparameters of DAS-like architecture optimization in a simple, sequential search space as well as in a non-sequential search space, which we both propose, where the latter is inspired by the search space proposed in Liu et al. (2019). We investigate two types of one-dimensional inverse problems which allow for extensive experiments for each setting in order to analyze the robustness of DAS.

We show that DAS can automatically find well performing architectures, if the search space is well preconditioned. Yet, our study also shows that the performance of DAS heavily depends on hyperparameter choices. Moreover, DAS shows a large variance for any set of hyperparameters, such that the suitability of parameters as well as the overall performance can only be judged when considering a large number of runs. This finding challenges the understanding of DAS as a one-shot method for NAS. Equally concerningly, we find that the estimated network performance using jointly optimized, shared weights is often not well correlated with the reconstruction ability of the final model after operation selection and re-training, i.e., the continuous relaxation in DAS seems to be quite loose. In particular, this makes the search for good hyperparameters by optimizing for the DAS training objective near-impossible. Hyperparameter optimization w.r.t. the final architecture performance is even more expensive and seems to increase the variance in the results even further. Yet, overall, our study also shows that DAS can successfully be applied to inverse problems. Specifically, it improves over competitive random search baselines by a significant margin, when the search space contains a variety of harmful and beneficial operations. This finding is crucial, since well preconditioned search spaces can be non-trivial to determine in novel applications.

## 2 Related Work

The introduction of DAS (Liu et al., 2019) proposed a novel paradigm in NAS area, in contrast to previously predominant black-box optimization methods. By relaxing the discrete operation choices in a network and thus allowing for gradient-based optimization, DAS has proven to be beneficial in the search of good neural architectures with limited search budget. Building on this pioneering work, NAS research has gained significant momentum for further improvements over the original DAS approach (Pham et al., 2018; Liu et al., 2018; Dong and Yang, 2019; Cai et al., 2019; Xie et al., 2019; Chen et al., 2019; Akimoto et al., 2019; Xu et al., 2020; He et al., 2020; Chen and Hsieh, 2020; Wu et al., 2021; Zhang et al., 2021). To narrow down the vast amount of literature, we focus here only on weight-sharing literature that is in line with our case study.

**Stability of DAS**. There are only few works investigating the stability of DAS w.r.t the so-called rank disorder (White et al., 2023) (the low correlation between the estimated performance of the one-shot model and the performance after retraining) and a poor test generalization of the found architecture (Zela et al., 2020; Xu et al., 2020; Chu et al., 2020; Chen and Hsieh, 2020; Li et al., 2020, 2021). RobustDARTS (Zela et al., 2020) tracks the dominant eigenvalue $\lambda_{\max}^{\alpha}$ of the Hessian during the architecture search and implements a regularization and early stopping criterion based on this quantity for a more robust DAS search. Chen and Hsieh (2020) pick up the relationship between the Hessian during the architecture search and the performance gap during search and evaluation time. They propose a perturbation-based regularization to smooth the validation loss landscape. Xu et al. (2020) find that only connecting partial channels into the operation selection leads to a regularized search to improve the stability. Chu et al. (2020) use a sigmoid activation for the architecture weights instead of softmax to eliminate unfair optimization regarding the skip-connection operation. Yang et al. (2020) analyze the contribution of each component in a NAS approach within the search

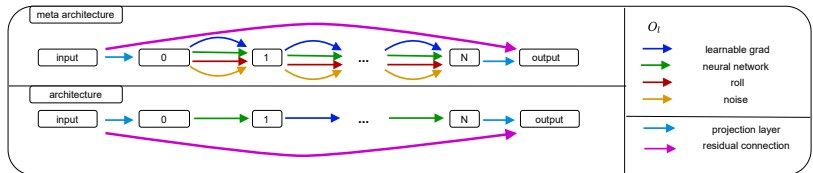

Figure 1: Investigated sequential meta-architecture. This setup is simple, yet it is able to represent DnCNN (Zhang et al., 2017)-like architectures.

space from Liu et al. (2019). They highlight that a performance-boosting training pipeline, often a result of expert knowledge, is more important for the evaluation of architectures than the search itself. Li et al. (2021) uses a single-level optimization to improve the poor test generalization. These findings motivate our analysis of the potential benefits of DAS in a setting different from image classification.

**Reconstruction.** Previous work on reconstruction of inverse problems via learned approaches has often focused on unrolled optimization schemes, such as unrolled PDHG in Riegler et al. (2016) and Adler and Öktem (2018). These architectures, also referred to as variational networks (Klatzer et al., 2016; Hammernik et al., 2017), are constructed by unrolling existing optimization routines that solve inverse problems and adding learning components in blocks which are either recurrent, as e.g. in Aggarwal et al. (2019) or fully independent as in Hammernik et al. (2018). In this investigation we will focus on parameterized gradient descent layers which can be seen as the most fundamental building block of these optimization routines.

## 3 Differentiable Architecture Search

We summarize DAS (Liu et al., 2019) in the following. To determine which operation $o^{(j)}$ is most suitable to be applied to the feature $x^{(j)}$, one defines a set of candidate operations $o_t \in \mathcal{O}, t \in \{1, \ldots, | \mathcal{O} | =: T\}$ where the NAS optimization problem follows the objective to select the optimal (discrete) arrangement of these operations in a neural architecture. DAS searches over the continuous relaxation of this discrete problem, with

$$o^{(j)} = \sum_{t=1}^{T} \beta_{o_t}^{(j)} o_t, \qquad \beta_{o_t}^{(j)} = \frac{\exp(\alpha_{o_t}^{(j)})}{\sum_{t'=1}^{T} \exp(\alpha_{o_{t'}}^{(j)})}, \tag{1}$$

where $\alpha = (\alpha_{o_t}^{(j)})$ are *architecture parameters* that determine the selection of exactly one candidate operation in the limit of $\beta$ becoming binary. Instead of looking for binary parameters directly, the optimization is relaxed to the soft-max of continuous parameters $\alpha$.

DAS formulates this search as a bi-level optimization problem in which both, the network parameters $\theta = \{\theta^{(j)}\}_{j=1}^{N}$ and the architecture parameters $\alpha$, are jointly optimized on the training and validation set, respectively, via

$$\min_{\alpha} \mathcal{L}_{val}(\theta(\alpha), \alpha) \qquad \qquad \text{s.t. } \theta(\alpha) \in \arg\min_{\theta} \mathcal{L}_{train}(\theta, \alpha), \tag{2}$$

where $\mathcal{L}_{val}$ and $\mathcal{L}_{train}$ denote suitable loss functions for the validation and training data. The optimization is done by approximating Eq. 2 (right) by one (or zero) iterations of gradient descent, and depends on several hyperparameters such as initial learning rates, learning rate schedules and weight decays for both architecture and model parameters. The final, discrete architecture is obtained by choosing the most likely operation $\hat{o}^{(j)} = \arg\max_{o_t} \alpha_{o_t}^{(j)}$ for each node. Subsequently, this final network is retrained from scratch. Thus, the fundamental assumption that justifies the idea of DAS is that the performance reached by the final network architecture on the validation

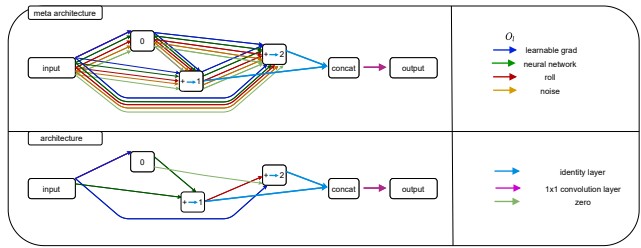

Figure 2: Investigated meta-architecture in the non-sequential search space.

set (the *architecture validation*) is highly correlated with the performance of the relaxed DAS approach obtained in Eq. 2 (the *one-shot validation*). Only then, the architecture found during DAS optimization can also be expected to perform well after retraining.

While the originally proposed method optimizes so-called *cells*, which are stacked in order to define the overall neural network architecture, and defines each cell in the form of a directed acyclic graph (DAG), we conduct most parts of our DAS study on a sequential, easy-to-interpret, meta-architecture to be described below (Fig. 1). To exclude that our findings are merely due to this special search space, we also consider experiments resembling the original setup in Liu et al. (2019). Our sequential architecture consists of $N$ nodes $x^{(i)}$, where $x^{(0)}$ represents the input data and the result $x^{(j+1)}$ of any layer is computed by applying some operation $o^{(j)}$ to the predecessor node $x^{(j)}$, i.e., $x^{(j+1)} = o^{(j)}(x^{(j)}, \theta^{(j)})$, where $\theta^{(j)}$ are the (learnable) parameters of operation $o^{(j)}$.

### 3.1 Proposed Search Spaces

Our following analysis of DAS for inverse problems will deliberately not be targeting settings that yield good results by design. In contrast, we propose two search spaces with different complexities that allow to analyze the stability and generalization performance of DAS under varying degrees of difficulty, in ascending order: (i) finding a good (linear) sequence of operations from meaningful choices of operations, (ii) finding a good (linear) sequence of operations where the set of operations to choose from contains good operations as well as harmful operations (the model needs to learn to avoid these), (iii) finding a good non-linear, acyclic computational graph of operations from meaningful choices of operations (this is the conventional DARTS setting), (iv) finding a good non-linear, acyclic computational graph of operations, where the set of operations to choose from contains good operations as well as harmful operations (the model needs to learn to avoid these).

Such search spaces allow to investigate the properties of DAS methods under various and realistic conditions. Specifically, not for all tasks, we can assume that the set of well-performing, beneficial operations is given or even complete. Therefore, it is desirable that methods perform reliable even if poor operation choices are available.

#### 3.1.1 Sequential Search Space.
For the simpler, sequential search space, we propose the meta-architecture shown in Fig. 1, which should be specifically well-suited for examples of signal recovery from known data formation processes such as blurring and subsampling with noise. From a pre-defined set of operations, we choose operations sequentially before adding the output to a residual branch. Image recovery networks such as DnCNN (Zhang et al., 2017) are contained in this meta-architecture. In practice, we search for 10 successive layers. As discussed above, the search space deliberately contains benign as well as harmful operations (see Sec. 3.1.3). This allows the evaluation of the effectiveness of DAS in any setting via the distinction of two cases: Training on all operations versus training only on beneficial operations. *A good architecture search algorithm should reliably find the optimal operations, even when presented with sub-optimal choices.*

#### 3.1.2 Non-Sequential Search Space.
For the non-sequential search space, we construct a cell structure with 5 states, and allow for arbitrary forward connections among the same set of operations as in

the sequential setting, but also for a {*zero*} operation, resulting in up to 15 operational connections. The outputs of the last two states are concatenated and flattened via a 1D convolution. We utilize two of these cells in succession, so that the depth of this search space with in total 10 nodes is comparable to our sequential search space. Fig. 2 visualizes an exemplary cell meta architecture during architecture search and the found cell architecture, which is then retrained. We choose in each cell one operation out of several as a connection between each node in the cell. This setting is more directly comparable to the original DARTS formulation (Liu et al., 2019), which contains a cell structure with multiple possible connections between sequential states.

**3.1.3 Network Operations for Inverse Problems.** In both the sequential and non-sequential setting, we search for the optimal architecture that can be defined using operations selected from a defined set $\mathcal{O}_l$. Specifically, we propose to use four operations, two of which are benign and potentially beneficial by design.

The first benign operation is motivated from rolled-out-architectures (see e.g. Gregor and LeCun (2010); Schmidt and Roth (2014); Kobler et al. (2017)) and tries to embed model-based knowledge about the recovery problems into the networks architecture. In this paper we consider problems which can be phrased as linear inverse problem, in which the quantity $x$ ought to be recovered from data $y = Ax + \text{noise}$ for a linear operator $A$. While the precise type of algorithm is typically dictated by (smoothness) properties of the regularization, a partially parameterized network-based approach has a lot of freedom to choose from template layers based on the inverse problem $y = Ax + \text{noise}$, i.e., $\arg\min_x D(Ax, y)$, where $D$ is a data formation term arising from the distribution of noise present, i.e., $D(Ax, y) = \frac{1}{2}||Ax - y||^2$ for Gaussian noise. This optimization objective yields templates such as a gradient descent layer: $x^{k+1} = x^k - \tau A^T \nabla_x D(Ax^k, y)$, for input $x^k$ and output $x^{k+1}$ of a new layer. The gradient layer can be turned into a learnable operation by introducing a learnable mapping $\mathcal{F}(x, \theta)$ after the gradient step, $x^{k+1} = x^k - \tau A^T \nabla_x D(Ax^k, y) - \tau \mathcal{F}(x^k, \theta)$ as a **learnable gradient descent layer** in our operation set $\mathcal{O}_l$.

The second benign layer is a fully-learned **neural network layer** in our operation set $\mathcal{O}_l$, that learns an appropriate mapping $\mathcal{F}(u, \theta)$ without knowledge of the operator $A$: $x^{k+1} = \mathcal{F}(x^k, \theta)$. For both layers, the learnable mapping $\mathcal{F}(x, \theta)$ is parametrized by a small convolutional network, consisting of a convolution layer, followed by batch normalization, ReLU and a second convolution layer.

These two layers, learnable gradient descent layer and neural network layer, are by design beneficial operations. To complement these beneficial layers we also include two potentially harmful operations to each operation set; a gradient layer with white Gaussian noise, **noise layer**, and a **roll layer**, which rolls the inputs in all dimensions. In total, we set $\mathcal{O}_l = \{$learnable gradient descent, 2-layer-CNN, roll, noise$\}$.

## 4 Evaluating DAS for Inverse Problems

In the following, we describe the experimental setting in which we evaluate DAS for inverse problems. Thereby, we focus on small problem instances to be able to evaluate the framework not once but in several runs such as to evaluate the statistics of the results. This setup also allows to gain insights on the dependence of DAS' performance on the chosen hyperparameters.

### 4.1 Experimental Setup

**Data Generation.** For a fast synthetic test we generate one-dimensional data sampling cosine waves of varying magnitude, amplitude and offset, and search for models to recover these samples from distorted measurements. We consider two distortion processes with varying difficulty: First, Gaussian noise and blurring and second, in addition to these, a subsampling by a factor of 4.

We generate these synthetic one-dimensional cosine data from $N = 50$ equally spaced points $\omega_i$ on the interval $[-\frac{\pi}{2}, \frac{\pi}{2}]$ with the model $x_i = \cos(f\omega_i + O_x) + O_y$ for a random frequency $f$

drawn uniformly from the interval $[0, 2\pi]$ and offsets $O_x$ and $O_y$ drawn from a normal Gaussian distribution. Such random drawn waves comprise our ground truth training data. We then generate measured data via the linear operation $A$ and addition of noise, i.e., $y = Ax + n$ with $n \in \mathcal{N}(0, \sigma_n)$. These pairs $(y, x)$ represent our training data. We sample new examples on-the-fly during both training and validation, so that no confounding effects of dataset size exist. All validation and training loss evaluations are each based on 2 432 randomly drawn samples. The performance of all models is evaluated in terms of their average peak signal to noise ratio (PSNR) on validation data. For all experiments we chose $\sigma_n = 0.01$. For the *blur* experiments, the linear operator $A$ is a Gaussian blur with kernel size 7 and $\sigma_b = 0.2$. For the *downsampling* experiments, this Gaussian blur is followed by a subsampling by a factor 4.

**Hyperparameter Optimization**. Our one-dimensional case study allows us to optimize DAS training hyperparameters with more granularity than it would be possible for image classification tasks. While we run our first experiments using manually tuned hyperparameters (see Appendix for details), we also consider the behavior and stability of DAS under optimized hyperparameters. We stress that we consider this mainly as an analysis tool - given that NAS itself is a hyperparameter optimization on which we stack another, and acknowledging that this optimization is practically intractable when larger problems are considered. To improve hyperparameters, we apply BOHB (Falkner et al., 2018), a Bayesian optimization method with hyperband (Li et al., 2018) and run BOHB for 128 hyperband iterations, which is an affordable budget in this one-dimensional data setting. Please note that BOHB is not an exhaustive search. Thus there are no guarantees for any optimality criteria on the found hyperparameters within a certain budget. Practically, we optimize the hyperparameters w.r.t. the one-shot validation performance, "BOHB-one-shot", and w.r.t. the final architecture performance, "BOHB". Note here that hyperparameter search that maximizes the final architecture performance instead of the one-shot validation performance is twice as expensive (on top of the already expensive hyperparameter search), due to the need for retraining.

## 4.2 Experimental Analysis

We first investigate the performance of DAS on the sequential search space. For our analysis of DAS as a one-shot method, we evaluate the statistics of the search over 75 trials of DAS as well as several baselines such as (i) setting all operations to *Learnable Grad.* or *Net* (i.e., learnable gradient descent or 2-layer CNN), (ii) picking a random architecture (random sampling), and (iii) performing a random search within an equal time budget as required by one DAS run. For the analysis, we distinguish between two different operation sets: only good operations (good ops.) and the complete operation set $\mathcal{O}_l$ (all ops.).

The results in Tab. 1 indicate that DAS works well for inverse problems. It proposes good architectures given the complete operation set for both considered data formation methods, *blur* and *downsampling*, resulting in architectures with a median PSNR of 18.57 and 16.12. Thereby, DAS also outperforms architectures consisting of only one good operation in both operations set cases, especially when considering the best found architecture using DAS. Practically, these experiments thus lead to a first interesting result for applied inverse problems: *The best found architecture is a hybrid version that mixes both beneficial operations, possibly suggesting that the best way to approach inverse problems are neither plain (convolutional) networks nor pure unrolling schemes.*

Next, we compare DAS to random sampling (random selection of the operation at each layer) and random search approaches. To allow for comparison at an equal time budget for the latter (random search in Tab. 1), we evaluate 5 randomly sampled architectures and report the best for each trial. One random evaluation using only good operations, i.e., the training of one random architecture, takes on average 57 sec. versus 2min. 39 sec. for one DAS run. For the sequential search space that purely consists of benign operations, random search outperforms DAS with a median PSNR of 22.75 versus 21.6 on *blur* and 17.56 versus 16.66 on *downsampling*. Thus in this scenario, *random search outperforms DAS when used as a one-shot model*. In addition, the simple

Table 1: Architecture validation PSNR values for 1D inverse problems. Shown is the maximal, mean, median, minimal and standard deviation of PSNR over 75 trials.

| | Method | Blur | | | | | Downsampling | | | | | |
|---|---|---|---|---|---|---|---|---|---|---|---|---|
| | | Architecture Val. (PSNR) | | | | | Architecture Val. (PSNR) | | | | | Runtime |
| | | Max. | Mean | Med. | Min. | Std. | Max. | Mean | Med. | Min. | Std. | min:sec |
| good ops. | Learnable Grad. only | 17.45 | 16.36 | 16.49 | 8.80 | 0.99 | 14.35 | 13.24 | 13.55 | 7.95 | 1.22 | 0:57 |
| | Nets only | 21.63 | 19.45 | 20.71 | 8.07 | 2.98 | 16.92 | 13.13 | 14.05 | 1.08 | 3.46 | 0:57 |
| | DAS | 23.46 | 21.56 | 21.60 | 17.60 | 1.03 | 18.03 | 16.36 | 16.66 | 11.48 | 1.18 | 2:39 |
| | Random Sampling | 24.04 | 22.05 | 22.17 | 19.43 | 0.77 | 18.10 | 16.73 | 16.84 | 14.51 | 0.79 | 0:57 |
| | Random Search | **24.04** | **22.60** | **22.63** | **21.29** | 0.48 | **18.10** | **17.40** | **17.49** | **16.26** | 0.45 | 2:55 |
| all | DAS | **22.86** | **15.64** | **18.57** | 7.95 | 6.09 | **18.01** | **15.39** | **16.12** | 1.04 | 3.13 | 2:53 |
| | Random Sampling | 20.86 | 9.45 | 8.10 | 7.88 | 2.85 | 13.78 | 5.08 | 4.31 | 1.11 | 2.95 | 0:35 |
| | Random Search | 20.86 | 12.12 | 12.24 | **8.09** | 4.11 | 13.78 | 9.61 | 9.77 | **6.22** | 2.12 | 2:55 |

Table 2: Architecture validation PSNR values for 1D inverse problems for the non-sequential search space. Shown is the maximal, mean, median, minimal and standard deviation of PSNR over 100 trials.

| | Method | Blur | | | | | Downsampling | | | | | |
|---|---|---|---|---|---|---|---|---|---|---|---|---|
| | | Architecture Val. (PSNR) | | | | | Architecture Val. (PSNR) | | | | | Runtime |
| | | Max. | Mean | Med. | Min. | Std. | Max. | Mean | Med. | Min. | Std. | min:sec. |
| good ops. | Learnable Grad. only | 13.19 | 12.41 | 12.38 | 11.81 | 0.23 | 11.30 | 8.89 | 9.59 | 1.13 | 2.60 | 7:33 |
| | Nets only | **16.35** | **14.83** | **15.82** | 8.01 | 2.65 | **13.63** | **13.07** | **13.06** | **12.53** | 0.18 | 6:35 |
| | DAS | 15.34 | 13.08 | 12.51 | 12.02 | 1.06 | 13.22 | 10.22 | 10.43 | 1.06 | 1.82 | 11:16 |
| | Random Sampling | 16.20 | 11.32 | 11.95 | 7.88 | 3.08 | 13.15 | 6.26 | 5.72 | 1.01 | 5.05 | 0:39 |
| all | DAS | **16.15** | **13.56** | **13.73** | 8.02 | 2.23 | 13.31 | **9.47** | **8.61** | 7.93 | 1.66 | 12:27 |
| | Random Sampling | 16.05 | 9.56 | 8.17 | **7.92** | 2.51 | **13.39** | 4.37 | 3.21 | 1.06 | 3.70 | 0:36 |

random sampling approach also outperforms DAS. This is different when harmful operations are added. However, for a search on the full operations set $\mathcal{O}_l$, DAS outperforms both simple random baselines. In addition, we can see, that the random search approach is able to find the best architecture in terms of minimal value, resulting here in stability against worst-case scenarios.

Since the original DAS formulation in Liu et al. (2019) contains a cell structure with multiple possible connections between sequential states, allowing for a larger degree of freedom in combining computational results, it is a-priori conceivable that some of the stability of DAS could be conferred through this structure. Therefore, we now analyze the DAS performance on the non-sequential DAS like search space exemplified in Fig. 2. Table 2 however shows that this wider search space does not improve the overall performance. Indeed the non-sequential search space hampers not only the DAS search significantly but also all other approaches, resulting in lower architecture performances for both data formations. In this setup, the *Nets only* baseline, that uses the 2-layer CNN for all operations, performs best. As above, we observe a significant drop in the performance of DAS when harmful operations are included in this non-sequential search space. In this case, as before, DAS can significantly outperform the random baseline but not reliably determine the obviously best operation. In contrast to the previous sequential search space, in which the random baseline (random search and sampling) shows stable behavior against worst-case scenarios, this does not hold in this non-sequential search space. Here, DAS shows a better stability, with less standard deviation. As the performance in this non-sequential search space is lower than in the sequential search space, we consider only the latter one in the following.

### 4.3 Correlation between Architecture and DAS Performance

Figure 3 plots the trials considered in Tab. 1, scattering the values of all trials separately with architecture performance (y-axis), which is computed after retraining the final architecture versus

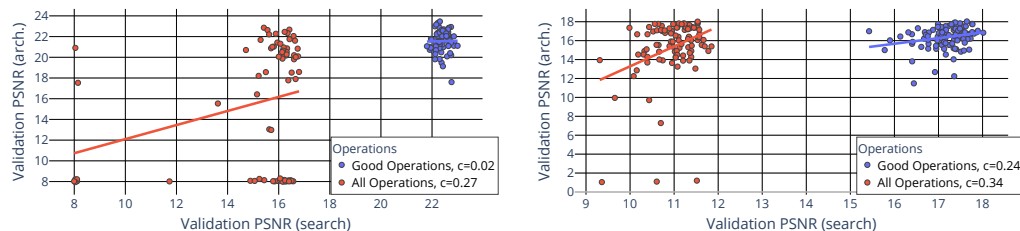

Figure 3: Top: Scatter plot corresponding to Tab. 1 showing architecture PSNR (y-axis) plotted against 1-shot validation PSNR (i.e., the validation performance on the DAS objective). Left: Blur. Right: Downsampling.

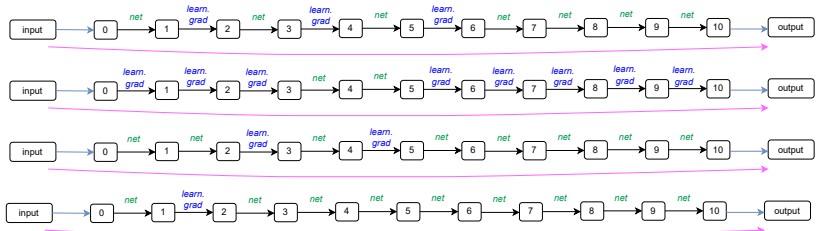

Figure 4: Found architectures using Random Sampling and DAS from Tab. 1. Top: Best found architecture using Random Sampling. Middle (first): Worst found architecture using Random Sampling. Middle (second): Best found architecture using DAS. Bottom: Worst found architecture using DAS.

the direct validation performance of the one-shot architecture (x-axis). We also plot a regression line over all trials and report the correlation of all trials in the legend, showing the linear fit has limited expressiveness. As discussed above, the correlation of these quantities is a fundamental assumption of DAS. However, this first experiment indicates a correlation problem: The assumption that a better *one-shot validation* implies a better *architecture validation* does not always hold.

These plots also show that DAS' behavior is highly problem-dependent: The *downsampling* dataset (right) shows that, although the mean value of DAS can be non-optimal, search performance and architecture performance are weakly correlated, even if the best architecture only has average search performance. The closely related *blur* dataset (left) shows an entirely different behavior with different "failure" cases, from which we can observe with the given hyperparameters that 1) either DAS proposes architectures with low (one-shot) search validation PSNR (i.e., it fails), or that 2) DAS works but does not predict a useful architecture (low architecture validation PSNR), or that 3) DAS does predict a useful architecture, but is unrelated to its search performance. Only the best proposed architectures perform well in both. To further analyze the correlation, we investigate DAS behavior under optimized training hyperparameters.

We visualize the best and worst found architecture in the sequential search space setting with only beneficial operations on *blur* in Fig 4. As we can see, random sampling's found best architecture iterates in the first 6 operations between *Net* and *Learn. Grad*, followed by only *Net*, while the best found architecture using DAS has less *Learn. Grad* operations but also stacks the *Net* operation at the last layers.

We evaluate DAS using 5 different training hyperparameter sets; two are chosen manually, H1 and H2 (H1 are the hyperparameters used in Sec. 4.2), whereas the other three are tuned using BOHB, as descibed in Sec. 4.1. In Tab. 3, we use BOHB to tune hyperparameters for the one-shot validation performance for both *blur* (BOHB-one-shot-Blur) and *downsampling* (BOHB-one-shot-DS), individually, and also to target the final validation performance for *blur* (BOHB-Blur). We

Table 3: Architecture validation PSNR values for 1D inverse problems with different hyperparameter sets. Shown is the maximal, mean, median, minimal and standard deviation of PSNR over 75 trials.

| | Hyperparameters | Blur | | | | | Downsampling | | | | |
|---|---|---|---|---|---|---|---|---|---|---|---|
| | | Architecture Val. (PSNR) | | | | | Architecture Val. (PSNR) | | | | |
| | | Max. | Mean | Med. | Min. | Std. | Max. | Mean | Med. | Min. | Std. |
| good ops. | H1 | 23.46 | 21.56 | 21.60 | 17.60 | 1.03 | 18.03 | 16.36 | 16.66 | 11.48 | 1.18 |
| | H2 | 23.46 | 21.43 | 21.63 | 13.23 | 1.53 | 18.20 | 16.57 | 16.78 | 13.96 | 0.92 |
| | BOHB-one-shot-Blur | 22.83 | 20.86 | 20.75 | **18.49** | 1.08 | **18.42** | **16.83** | **16.95** | **15.21** | 0.71 |
| | BOHB-one-shot-DS | 22.33 | 20.65 | 20.96 | 8.58 | 1.79 | 17.51 | 15.33 | 15.84 | 1.17 | 2.73 |
| | BOHB-Blur | **23.57** | **22.05** | **22.38** | 8.04 | 1.87 | 18.26 | 14.63 | 15.93 | 1.11 | 4.47 |
| all | H1 | 22.86 | 15.64 | 18.57 | **7.95** | 6.09 | 18.01 | 15.39 | 16.12 | 1.04 | 3.13 |
| | H2 | **23.10** | **16.77** | **19.88** | 7.86 | 5.69 | 17.82 | **15.93** | **16.21** | **7.27** | 1.66 |
| | BOHB-one-shot-Blur | 22.47 | 15.57 | 18.04 | 7.90 | 5.59 | 17.73 | 14.36 | 14.57 | 6.06 | 1.73 |
| | BOHB-one-shot-DS | 22.41 | 14.43 | 14.41 | 4.36 | 5.88 | **18.12** | 12.36 | 13.65 | 1.09 | 3.73 |
| | BOHB-Blur | 22.94 | 12.76 | 8.21 | 7.91 | 6.17 | 17.91 | 15.04 | 15.44 | 1.09 | 2.63 |

Figure 5: Scatter plot corresponding to Tab. 3 with BOHB-optimized hyperparameters, showing architecture PSNR (y-axis) plotted against one-shot validation PSNR (x-axis). Left: Blur with hyperparameters (top) *BOHB-one-shot-Blur* and (bottom) *BOHB-Blur*. Right: Downsampling with (top) *BOHB-one-shot-DS* and (bottom) hyperparameters *BOHB-Blur*.

also plot the results for all BOHB found hyperparameter trials in Fig. 5. The plot shows that the correlation for both data formation methods increases with the corresponding BOHB-one-shot tuned hyperparameters, with also a higher range of the search validation PSNR, as expected. This experiment also shows a rather surprising outcome: In the case of *blur*, the average performance is on par with the manually chosen hyperparameters H1 and H2, whereas the performance for *downsampling* decreases, especially when all operations are considered. In addition, the best architecture PSNR over 75 trials decreases in both cases using the dataset specific hyperparameters. Overall, the apparent stabilization via optimization of the search loss removes not only negative, but also positive outliers. Furthermore, hyperparameters optimized for one dataset do not transfer well to the other. Using BOHB to target the final validation performance for *blur* (BOHB-Blur) instead of the one-shot validation performance has also a positive impact on the one-shot validation and architecture validation correlation, compared to the manually chosen hyperparameters H1 and H2 in Fig. 3.

Table 4: Architecture validation PSNR values for BSDS Blur reconstruction for the sequential search space. The maximal, mean, median, minimal, and standard deviation of PSNR over 3 trials for only good operations are shown.

| Method | Validation PSNR (eval) | | | | | Runtime |
|---|---|---|---|---|---|---|
| | Max. | Mean | Median | Min. | Std. | h:min:sec |
| DnCNN (Nets only) | 25.96 | 25.94 | 25.93 | 25.92 | 0.02 | 30:41 |
| DAS | 25.95 | 25.91 | 25.93 | 25.85 | 0.05 | 1:50:02 |
| Random Sampling | **26.09** | **26.02** | **26.04** | 25.93 | 0.08 | 20:03 |

In conclusion, we find two schools of thought when evaluating the performance of DAS. For maximal performance, we should *not understand DAS as a one-shot search approach*, but as a component in a *larger search that proposes trial architectures*. For average performance, and immediate performance with a single DAS run, we should be optimizing the search performance and maximize its correlation with architecture performance - although as our experiments show, this is non-trivial even when searching for these hyperparameters in an automated fashion. We stress that the two directions are not at odds with each other, yet problems can arise in the literature when comparing proposed improvements of DAS across both. Some algorithmic improvements of DAS are more likely to improve best-case performance, whereas others are more likely to impact single trial stability, and both can not be directly compared.

### 4.4 Image Reconstruction Experiments

In this section, we evaluate DAS on the Berkeley Segmentation Dataset and Benchmark (BSDS300) (Martin et al., 2001) for image denoising in the sequential search space using the good operation set. We compare DAS to the random sampling baseline as in the previous 1D experiments and to the DnCNN (Zhang et al., 2017) baseline. The latter architecture only contains the operation *Net*. Tab. 4 shows the results for three runs. The random sampling baseline improves over DAS and the DnCNN baseline. DAS found in 2 our of 3 runs an architecture containing only *Learnable Grad*. We also see that a hybrid version that mixes both beneficial operations improves over both plain networks. As it is apparent from the compute times in Tab. 4, this setting can not be the basis of an exhaustive study as we provide it in the 1D case. Yet it confirms the trends that we observe therein.

## 5 Conclusions

In this paper we analyzed DAS on one-dimensional inverse reconstruction problems. We show that DAS improves over a random search baseline by a significant margin, especially if the available set of beneficial operation is not determined in advance. Further, we make the following findings: While it is possible to find well-performing architectures using DAS, multiple runs of the same setting yield a high variance, challenging the common understanding of DAS as a one-shot method. Moreover, the ability to find well-performing architectures is highly dependent on the specific choice of hyperparameters. Therefore, we emphasize for the future the necessity to (1) look at a full statistical evaluation of DAS performances over multiple trials, (where-ever affordable), and (2) show a correlation between the search and final architecture performances for any method that reports improved results based on a more faithful minimization of the one-shot DAS objective.
**Limitations** In this study we evaluated the commonly used DAS approach on inverse problems w.r.t hyperparameter sensitivity. This study is limited to 1D data, which however makes the analysis and especially the hyperparameter optimization affordable. The next step would be to analyze this effect in larger computer vision settings, in particular investigating the influence of the proposed search space on the test generalization using bi-level optimization, without the need to change it to single-level approaches.

**Broader Impact:** After careful reflection, the authors have determined that this work presents no notable negative impacts to society or the environment. It rather aims to contribute to sustainable progress in NAS.

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

## A  Non-Degenerated Search

Here, we visualize in Fig. 6 the trials without degenerated search results from the experiment in Tab. 1, i.e., search PSNR results with less than 9 PSNR, and without generated results in total, i.e., search and eval PSNR less than 9 PSNR. As we can see, the correlation decreases, if we only remove the degenerated search outcome but improves as soon as all degenerated results are removed.

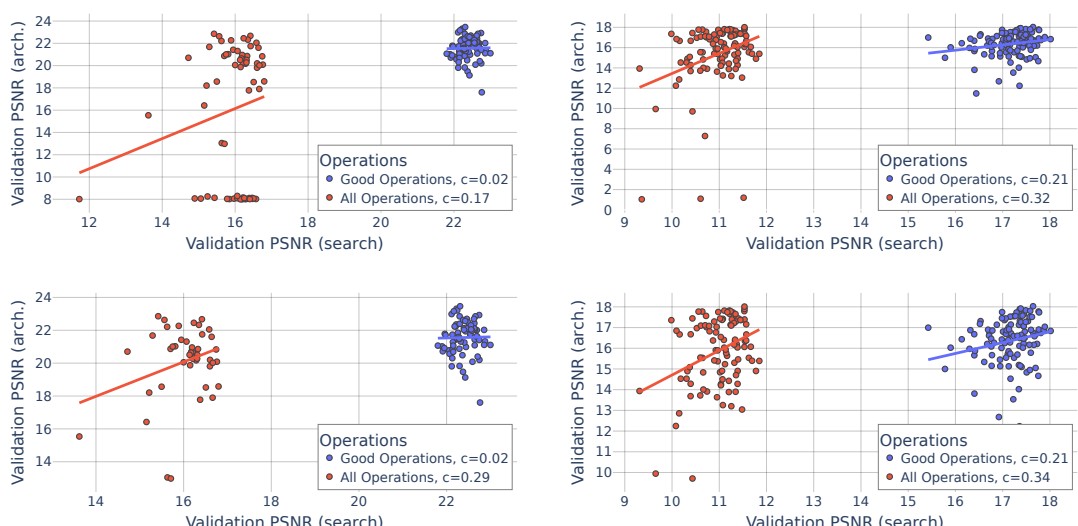

Figure 6: Scatter plot corresponding to Tab. 1 **without degenerated results** showing architecture PSNR (y-axis) plotted against 1-shot validation PSNR (i.e., the validation performance on the DAS objective) for *blur* (left) and *downsampling* (right). Top: Scatter plots without degenerated search results. Bottom: Scatter plots without all degenerated results.

## B  Improving the Initialization

Several works, such as Zela et al. (2020); Li et al. (2021), investigate the instability of the bi-level approximation of DAS w.r.t. the weight initialization; the random initialization of the network weights can cause promising operations having poor initialization and thus tend to be entirely discarded during the architecture search, which eventually leads to poor test generalization. Li et al. (2021) overcomes this by proposing a single-level optimization. Therefore, we evaluate the impact of this initialization by modifying the DAS search, such that it only has to search for the optimal architecture parameters to build the resulting architecture, also resulting in a single-level search approach.

For this *DAS-single* approach, we pre-train the operations {learnable gradient descent} and {2-layer-CNN} as baseline architectures, compared to *Learnable Grad. only* and *Nets only* from Sec. 4.2, and keep the operation weights fixed. This is generally only possible for the feed-forward architectures that we consider and requires only a weak specialization between layers. Thereby,

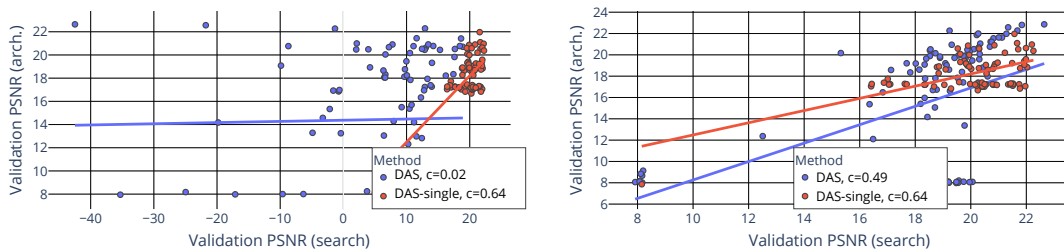

Figure 7: Left: Architecture search with BOHB-searched hyperparameter sets for DAS with single-level optimization on the blur data formation. Right: Both methods with their own BOHB hyperparameters on the blur data formation.

we avoid the random initialization of the operations weights in the DAS search. Figure 7 shows the results of DAS-single search with BOHB-optimized hyperparameters for all operations. Notably, BOHB-optimized hyperparameters for the DAS-single one-shot validation (Fig. 7 left) lead to a positive impact on the correlation of the one-shot and architecture validation PSNR using DAS-single and to a negative impact for DAS. In addition, when comparing DAS and DAS-single with their hyperparameters being individually optimized with BOHB with respect to their one-shot validation (Fig. 7 right), DAS finds a higher architecture validation PSNR than DAS-single, whereas DAS-single becomes more robust against possible outliers, making this search less sensitive.

## C  Hyperparameter Stabitily in the Non-sequential Search Space

In this section we investigate the stability of our DAS framework w.r.t. hyperparameters within the non-sequential search space from subsubsection 3.1.2 for the *blur* data formation. We also visualize architectures found by our DAS approach for two different operation sets (Figure 11).

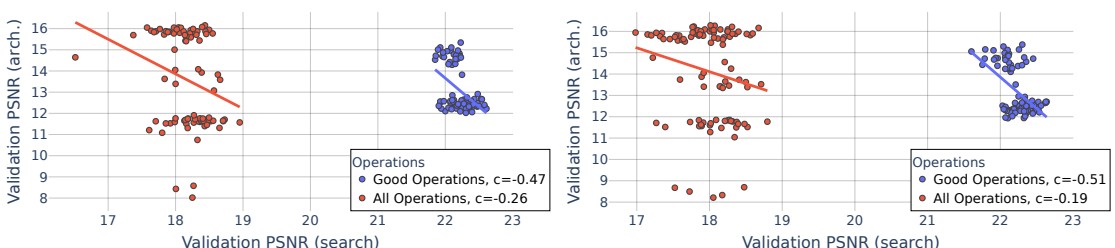

Figure 8: Scatter plot for the non-sequential DAS search space corresponding to Table 5, with hyperparameters *H1* (left) and *H2* (right), showing architecture PSNR (y-axis) plotted against 1-shot validation PSNR (i.e., the validation performance on the DAS objective).

We conduct experiments using the same BOHB-optimized hyperparameters as in subsection 4.3 and additionally included BOHB-optimized hyperparameters for this non-sequential search space for first targeting the one-shot validation performance (BOHB-Non-Seq-one-shot-Blur) and second targeting the final architecture performance (BOHB-Non-Seq-Blur). Table 5 however shows similar results as in subsection 4.3: changing the hyperparameters does not improve the stability of the search process. Figure 8 shows all trials for the non-sequential search space for the manually chosen hyperparameters *H1* and *H2*. This plot clarifies further that the search space change does not improve the DAS search process. The correlation between the one-shot validation and the architecture validation even becomes negative. Yet, these plots also show 2 different "failure" cases for both operations sets, only beneficial operations and all operations, and both data formations:

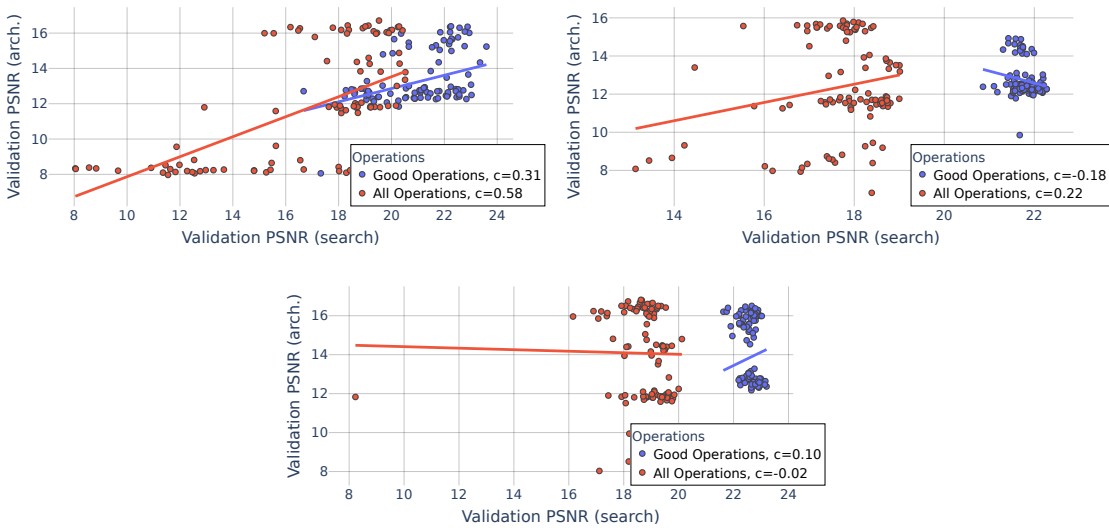

Figure 9: Scatter plot for the non-sequential DAS search space on blur with BOHB-optimized hyperparameters for this search space, showing architecture PSNR (y-axis) plotted against one-shot validation PSNR (x-axis). Top (Left): Blur with hyperparameters *BOHB-one-shot-Blur*. Top (Right): Blur with hyperparameters *BOHB-one-shot-DS*. Bottom: Blur with hyperparameters *BOHB-Blur*

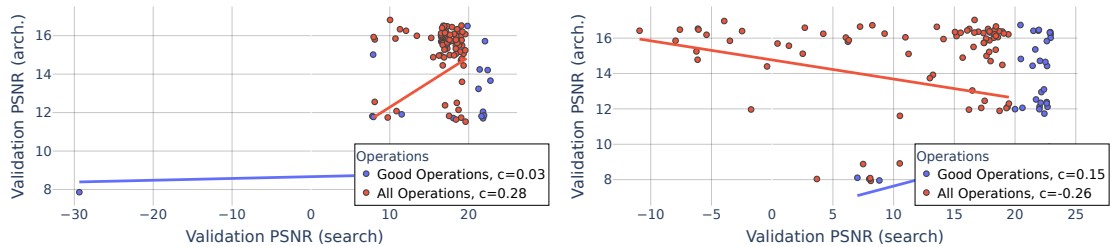

Figure 10: Scatter plot for the non-sequential DAS search space on blur with hyperparameters searched for this search space, showing architecture PSNR (y-axis) plotted against one-shot validation PSNR (x-axis). Left: Blur with hyperparameters *BOHB-Non-Seq-one-shot-Blur* Right: Blur with hyperparameters searched for the final architecture performance *BOHB-Non-Seq-Blur*.

The validation PSNR is stable, whereas the architecture validation performance is clustered in two different regions, one being very low and the other being around 15 PSNR. Note, the mean architecture validation PSNR for all operations in the sequential search space from subsection 4.3 in Table 3 is also around 15 PSNR.

For additional visualization, we also display the results using BOHB found hyperparameters in the sequential search space in Figure 9 as well as BOHB-found hyperparameters tuned for this non-sequential search space in Figure 10. However, hyperparameter search for the non-sequential search space via BOHB on both the one-shot validation performance and the architecture performance as a target does not improve the stability of the search, as demonstrated in Figure 10. Accordingly, we find on the one hand that the findings in the previous section 4.2 regarding the non-applicability of DAS as a one-shot model for inverse problems translate to a cell-based search space and on the other hand (investigating the overall performance metrics for both search spaces), that the

Table 5: Architecture validation PSNR values in the non-sequential search space for the 1D inverse problems setting with cosine data. Shown is the maximal, mean and median PSNR over 75 trials.

| Data | Hyperparameters | Architecture Validation (PSNR) | | | | | |
| | | Good Ops. | | | All | | |
| | | Max. | Mean | Med. | Max. | Mean | Med. |
|------|-----------------|------|------|------|------|------|------|
| | H1 | 15.34 | 13.08 | 12.51 | 16.15 | 13.56 | 13.73 |
| | H2 | 15.38 | 13.17 | 12.52 | 16.28 | **14.11** | **15.58** |
| | BOHB-one-shot-Blur | 16.38 | 13.25 | 12.76 | 16.71 | 11.73 | 11.8 |
| Blur | BOHB-one-shot-DS | 14.93 | 12.73 | 12.44 | 15.86 | 12.37 | 11.72 |
| | BOHB-Blur | 16.5 | **13.83** | **13.06** | 16.82 | 14.07 | 14.44 |
| | BOHB-Non-Seq-one-shot-Blur | 16.50 | 8.84 | 8.09 | 16.82 | 13.96 | 15.50 |
| | BOHB-Non-Seq-Blur | **16.74** | 9.73 | 8.11 | **17.03** | 13.45 | 15.42 |

sequential search space appears to be a helpful prior for architecture search for inverse problems, given that its PSNR scores are overall higher.

## D Visualizations

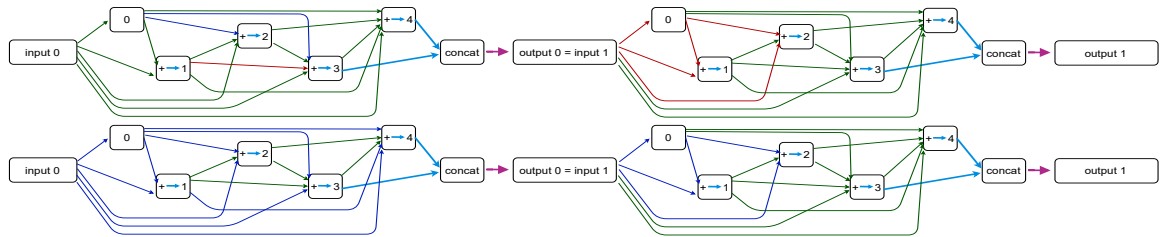

Figure 11: Found architectures in the non-sequential search space for two different operation sets for the data formation blur. Hyperparameter H1 is used for these searches. Top: all operations. Bottom: only beneficial operations.

In this section, we visualize in Figure 11 two found architectures using the H1 hyperparameters for the operation sets "all operations" and "only good operations" for the data formation blur in the non-sequential search space from the experiments in subsection 4.2.

## E Hyperparameters

In this section, we show the hyperparameters used for our experiments in the main paper. In Table 7 are the manually chosen hyperparameters H1 and H2 from Table 3 in the main paper. In addition Table 8 lists all BOHB optimized hyperparameters for the data formations blur and downsampling as well as the hyperparameters optimized for the final architecture performance and also for the DAS-single method; the search range for the BOHB search is given in the second column. In Table 9 the BOHB search hyperparameters for the non-sequential search space from Appendix C are listed. Table 6 lists all other general hyperparameters used for our experiments.

Table 6: General Hyperparameters.

| Hyperparameter | Default Value |
|----------------|---------------|
| Epochs | 50 |
| Batch size | 128 |
| Noise Level | 0.10 |

Table 7: Manually chosen hyperparameters H1 and H2

| Hyperparameter | H1 | H2 |
|---|---|---|
| Param. learning rate | 0.001 | 0.001 |
| Param. weight decay | $1e-8$ | $1e-8$ |
| Param. warm up | False | False |
| Alpha learning rate | 0.001 | 0.0001 |
| Alpha weight decay | 0.001 | 0.0001 |
| Alpha warm up | True | True |
| Alpha scheduler | Linear | Linear |
| Alpha optimizer | Gradient Descent | Gradient Descent |

Table 8: BOHB optimized hyperparameters for different data formations, objectives and methods.

| Hyperparameter | Search Range | BOHB-one-shot-Blur | BOHB-one-shot-DS | BOHB-Blur | BOHB-DAS-single |
|---|---|---|---|---|---|
| Param. learn. rate | $[1e-05, 1]$ | 0.0014232405 | 0.0020448382 | 0.0020882283 | 0.0014232405 |
| Param. weight decay | $[1e-08, 0.1]$ | $8.616e-07$ | $5.04e-08$ | $4.4e-08$ | $8.616e-07$ |
| Param. warm up | [True,False] | False | True | False | False |
| Alpha learn. rate | $[1e-05, 0.1]$ | 0.0836808765 | 0.0100063746 | $8.43195e-05$ | 0.025012337102395577 |
| Alpha weight decay | $[1e-05, 0.1]$ | $5.05099e-05$ | 0.0058022776 | 0.0127425783 | $1.390640076980444e-05$ |
| Alpha warm up | [True, False ] | False | True | True | False |
| Alpha scheduler | [None, Linear] | Linear | Linear | Linear | None |
| Alpha optimizer | [Adam, Gradient Descent] | Adam | Gradient Descent | Adam | Adam |

# F Computational Setup

All experiments in the main body were run on a single `Nvidia GTX 2080ti` graphics card of which two were utilized. The hyperparameter tuning with BOHB was conducted on a single `Nvidia GTX 1080 Ti` graphics card.

Table 9: BOHB optimized hyperparameters for the non-sequential search space for data formation blur and different objectives.

| Hyperparameter | Search Range | BOHB-Non-Seq-one-shot-Blur | BOHB-Non-Seq-Blur |
|---|---|---|---|
| Param. learn. rate | $[1e-05, 1]$ | 0.0050969066 | 0.0037014752 |
| Param. weight decay | $[1e-08, 0.1]$ | $2.423e-07$ | $1.4573e-06$ |
| Param. warm up | [True,False] | False | False |
| Alpha learn. rate | $[1e-05, 0.1]$ | $1.32499e-05$ | 0.0012395056 |
| Alpha weight decay | $[1e-05, 0.1]$ | 0.0010171142 | 0.0002855732 |
| Alpha warm up | [True, False ] | False | False |
| Alpha scheduler | [None, Linear] | None | None |
| Alpha optimizer | [Adam, Gradient Descent] | Adam | Adam |

