# OpenReview forum: "Differentiable Architecture Search: a One-Shot Method?"
_automl.cc/AutoML/2023/Conference — AutoML 2023 Workshop_

### Official Review · Reviewer_4Z5E · 2023-04-11

**Potential Impact On The Field Of Automl Rating:** 4
**Technical Quality And Correctness Rating:** 4
**Clarity Rating:** 4
**Actions Required To Increase Overall Recommendation:** Scale the problem to at least MNIST l…

**Summary Of Contributions:**

The paper presents a methodical evaluation of the Differentiable Architecture search (DAS). It uses a simple 1-D inverse problem as a tractable and unbiased problem for the evaluation of the effectiveness of DAS as an AutoML technique and also specifically as a  one-shot solution. The paper presents empirical studies which provide practically useful insights into DAS as a generic AutoML technique.
It uses Sequential and non-sequential search spaces to further understand the impact of search space design.
Paper reveals the effectiveness of DAS in general and its limitation as single-shot solution based on empirical evidence.

**Clarity:**

The paper is well written. It is easy to follow and the description of experiments and discussion of the results is very clear.

**Overall Review:**

It is a well-written paper on a very important topic. There are many AutoML techniques currently available for NAS problems, with no clear choices for specific problems. This paper attempts at understanding the issues for DAS algorithms at fundamental level. More of such studies (in breadth and scale) are needed to make progress in the field.

**Potential Impact On The Field Of Automl:**

The paper provides an entry into the methodical evaluation of DAS techniques. These studies can be scaled up to bigger benchmarks. Such studies will help to improve the course of AutoML techniques and also provide a reproducible "rules of thumbs" for applying AutoML to practical problems.

**Review Confidence:**

4: You are confident in your assessment, but not absolutely certain. It is unlikely, but not impossible, that you did not understand some parts of the submission or that you are unfamiliar with some pieces of related work.

**Review Rating:**

9: Strong Accept: Technically flawless paper with major impact and strong evaluation, with no obvious flaws. Should be highlighted in the program.

**Review Summary:**

-do-

**Technical Quality And Correctness:**

The experimentation is thorough and discussion of result bring high confidence to the results. It has tried to cover as many co-founding factors as I could imagine. eg. use of HPO techniques is reasonable (as opposed to exhaustive) which brings credibility to results.
The only understandable limitation is the lack of scale of datasets (results on limited 1-D space and image space may vary quite significantly), which may be tackled in further studies.

---

> ### Author Response · Authors · 2023-05-02
> **Response to Reviewer**
>
> We appreciate the reviewer's acknowledgment of the usefulness and importance of our study to investigate DAS on a more fundamental level.
>
> **Scale to image datasets.**
>
> We additionally evaluated DAS on the Berkeley Segmentation Dataset and Benchmark (BSDS300) for image denoising in Sec. 4.4 in our updated paper. We compare DAS to the DnCNN architecture (Zhang et al., 2017) and to a random sampling baseline, where the latter random sampling approach improves over both, DAS and DnCNN. Important here is, that the 1D setting allows for this exhaustive study, which is not easily feasible with larger image dataset.
>
>
> We thank the reviewer again for the input.

---

### Official Review · Reviewer_wSGU · 2023-04-12

**Potential Impact On The Field Of Automl Rating:** 2
**Technical Quality And Correctness:** The included technical aspects are so…
**Technical Quality And Correctness Rating:** 2
**Clarity Rating:** 3

**Summary Of Contributions:**

This work identifies the problem that differentiable architecture search (DAS) has mainly been studied in a very specific architectural search space for image classification. To understand how it realistically performs in a new domain, it is applied to simulated inverse problems with either a sequential or an interconnected structural space, either a good or a mixed-quality operator space, and various hyperparameter optimization strategies.

**Actions Required To Increase Overall Recommendation:**

* Extend the analysis of already-run experiments, including computation cost and analysis of DAS architectures
* Include a realistic dataset


**Clarity:**

Some concepts could be further explained, but the language is otherwise clear overall.

**Overall Review:**

**Positive aspects:**

* You did well to search across hyperparameter optimization settings, from hand-selected to various approaches of transferring hyperparameters or searching on just the search phase or also the retraining phase.

**Negative aspects:**

* What do the searched architectures actually look like? Only two such architectures are shown (and only in the supplementary material), but looking at trends of architectures across trials could help explain why DAS can't outperform baselines when only good operations are used. For example, what do the architectural parameter patterns look like over the course of search?

* Why do you think random search and random sampling got the same max scores for all combinations of data generation methods and operation spaces? This seems like a red flag and should be explained, in addition to clarifying the difference between these two random baselines. From my understanding, random sampling selects and trains a single model per trial, while random search uses the same compute time as DAS to repeatedly train random architectures?

* In Figure 3, the outliers seem to greatly affect the regression line and correlation score: maybe also add the line and score for the non-degenerate data points (or at least those without a degenerate search model score, since these models could in theory be pruned out)?

* Make sure to include some measure of variance in your result tables. These can be qualitatively estimated from the plots, but standard practices in DAS are to report maximum, mean, and standard error of performance.

* What is the scale and intuitive meaning of PSNR? This measure is not often used in AutoML papers, so describing it would help the audience interpret the results.

* The only datasets are single dimensional and generated. The results would be strengthened by also using a more realistic dataset.

* Computational cost is only mentioned in Table 1; total costs are not reported (although we can guess they are pretty minimal based on Table 1). 75 to 100 repetitions seems rather excessive: generally for AutoML 5-10 runs (that only differ by random seed) suffice, so you could reduce the number of repetitions in order to perform more variations of experiments, such as another dataset as suggested previously.



**Minor comments:**

* Line 83: "smoothen" -> "smooth"

* Line 169: add a paragraph break between "...beneficial by design." and "The first benign..."

* Line 186: add a paragraph break between "...convolution layer." and "These two layers..."

* FIgures 1 and 2 seem to have been stretched horizontally. It would be better to regenerate these plots with the desired aspect ratio so that it is easier to read.

* Table 3 caption "hyperparameter." -> "hyperparameter sets."

**Potential Impact On The Field Of Automl:**

This paper has a small impact on the field of AutoML. Although it novelly investigates DAS in a new domain of inverse problems, it is a rather minimal study that does not provide many new findings beyond other DAS studies. To increase impact and citability, the authors could further analyze the experiments already run as well as expand the experiments with realistic datasets.

**Review Confidence:**

4: You are confident in your assessment, but not absolutely certain. It is unlikely, but not impossible, that you did not understand some parts of the submission or that you are unfamiliar with some pieces of related work.

**Review Rating:**

6: Borderline Leaning Accept: Technically sound paper where reasons to accept outweigh reasons to reject. Please use sparingly.

**Review Summary:**

This paper provides a good preliminary study on how DAS may realistically perform in a new domain not as extensively studied as image classification. However, the provided studies are only conducted on generated toy datasets, and there are few novel claims regarding problems with DAS beyond those already identified for the image classification domain in other DAS works.

Post-rebuttal update: I have raised my score from 5 to 6.

---

> ### Author Response · Authors · 2023-05-02
> **Response to Reviewer**
>
> We appreciate the reviewer’s recognition of our work and the suggestions made.
>
> **1. What do the searched architectures actually look like?**
>
> We added visualizations of the best and worst found architecture by DAS and random sampling for blur and only beneficial operations in Fig. 4. As we can see, random sampling's found best architecture iterates in the first 6 layers between *Net* and *Learnable Grad*, followed by only *Net* layers, while the best-found architecture using DAS has less *Learnable Grad* operations but also stacks the *Net* operation at the last layers.
>
> **2. Why do you think random search and random sampling got the same max scores for all combinations of data generation methods and operation spaces?**
>
> Random search is an extended random sampling approach, which searches for several architectures with the same computing time as one DAS run. It does not train a random architecture repeatedly. We reran these experiments and updated the values.
>
> **3. In Figure 3, the outliers seem to greatly affect the regression line and correlation score: maybe also add the line and score for the non-degenerate data points.**
>
> We included new plots in Appendix A without the degenerated results (search results, and both search and eval results). These plots show that the correlation decreases if only degenerated search outcomes are removed but increases as soon as all degenerated results are removed.
>
> **4. Make sure to include some measure of variance in your result tables.**
>
> Thank you for pointing that out. We included the minimal value and the standard deviation for each experiment in the updated paper.
>
> **5. What is the scale and intuitive meaning of PSNR? This measure is not often used in AutoML papers, so describing it would help the audience interpret the results.**
>
> The Peak signal-to-noise ratio (PSNR) is based on MSE and is a logarithmic measurement for reconstruction quality. The higher the PSNR the better is the reconstruction.
>
> **6. The only datasets are single dimensional and generated. The results would be strengthened by also using a more realistic dataset.**
>
> We additionally evaluated DAS on the Berkeley Segmentation Dataset and Benchmark (BSDS300) for image denoising in Sec. 4.4 in our updated paper. We compare DAS to the DnCNN architecture (Zhang et al., 2017) and to a random sampling baseline, where the latter random sampling approach improves over both, DAS and DnCNN. However, we want to point out here that our extensive study including unbiased hyperparameter search is possible, since the search and evaluation times on the 1D data are rather small. Such a study is not easily feasible for larger datasets.
>
>
> **7. Computational cost is only mentioned in Table 1; total costs are not reported (although we can guess they are pretty minimal based on Table 1). 75 to 100 repetitions seems rather excessive: generally for AutoML 5-10 runs (that only differ by random seed) suffice, so you could reduce the number of repetitions in order to perform more variations of experiments, such as another dataset as suggested previously.**
>
> We included computational costs for the other experiments in Tab. 2, Tab. 3, and Tab. 4.
> Indeed in the general AutoML setting only 5-10 runs are conducted, given rather large computation costs. In this paper, we want to analyze the convergence behavior of DAS over several trials without being biased to different better-performing seeds. Especially, since the computational costs are small, we are able to conduct this large amount of trials to investigate the convergence behavior.
>
> Thank you for pointing out the typos in the minor comments, we changed them.
>
> We thank the reviewer again for the input and hope we addressed all remaining concerns.

---

> > ### Comment · Reviewer_wSGU · 2023-05-05
> > **Response to Authors**
> >
> > Thank you for your updates and responses. While most of the "negative aspects" were addressed, there is still the overarching problem that this paper doesn't have many novel claims or insights beyond existing DAS papers. I will raise my score slightly.

---

> > > ### Comment · Area_Chair_zWC4 · 2023-05-06
> > > **question for authors**
> > >
> > > Dear authors,
> > > Reviewer wSGU's main question here, also mentioned by reviewers GfZK and PEcP, is that of what readers should learn from your paper that they didn't already know from previous papers. Specifically, it seems to me that these 3 reviewers (and perhaps many other readers too) believe the problems you highlight are already known. While you demonstrate the problems in the scenario of toy inverse tasks more thoroughly than before, is there any sense in which toy inverse tasks are representative of the problems readers may care about? For example, do the performance of search algorithms on toy inverse tasks correlate with their performance on important tasks?
> > > AC

---

> > > > ### Author Response · Authors · 2023-05-07
> > > > **Contributions**
> > > >
> > > > Dear AC,
> > > >
> > > > thank you for specifying the remaining concerns on our paper.
> > > >
> > > > From the "inverse problems" perspective, our paper contributes the **first extensive analysis of the performance of differentiable architecture search (DAS)**. We can show that (unlike SotA methods suggest) hybrid architectures containing learnable gradient and net operations have the potential to outperform architectures that purely consist of either learnable gradient or net operations. We also show that, in this particular (yet commonly addressed) application, there is little benefit in considering non-sequential search spaces. This observation is of high practical relevance for the design of architectures for inverse problems. Our newly added experiments on BSDS 300 confirm these findings **beyond toy experiments**.
> > > >
> > > > From the "differentiable architecture search" perspective, several of our findings have been suspected to some extent to the community. E.g., we know that DAS can be outperformed and retraining of the found architecture is required to achieve good performance. Yet, existing literature focuses on well-established search spaces, that mostly contain well-performing architectures and for which overall well-performing hyperparameters are known. As a result, the fluctuation of DAS results can only be observed to a limited extent in this scenario. The application in a different setting allows us to observe DAS under less controlled conditions: neither the search space nor suitable hyperparameters are given a priori. Please consider that this is a far more realistic setting than the classical "image classification in CIFAR or ImageNet" setting for which the community has been optimizing hyperparameters for years. As a result, we can make, for the first time, **statistically significant observations of the behavior of DAS under varying (yet optimized) hyperparameters**. To our understanding, the results are striking and challenge the understanding of DAS as a one-shot method altogether.
> > > >
> > > > As a consequence, we conclude that DAS approaches should not only report results for several runs including std errors but also that the correlation between search validation accuracy and final validation accuracy has to be monitored. If this correlation is weak in a specific setting, the entire DAS paradigm is prone to failure. Yet, such analysis is **lacking is SotA DAS** publications. Our paper aims to create awareness to this issue in the community.
> > > >
> > > > Last but not least, we argue that, from a research perspective, even such aspects that have been suspected by the community (e.g. DAS is not always stable), have to be validated in a statistically significant way. Our setting allows a statistically significant analysis of DAS in a simple and computationally manageable setting and thereby closes this important gap.
> > > >
> > > > We hope that our explanation could address your remaining concerns are are happy for any further discussion!
> > > >
> > > > Best regards,
> > > > authors of submission #25

---

### Official Review · Reviewer_PEcP · 2023-04-13

**Potential Impact On The Field Of Automl Rating:** 2
**Technical Quality And Correctness:** 1. This paper has a very interesting …
**Technical Quality And Correctness Rating:** 2
**Clarity:** 1. The organization of this paper is …
**Clarity Rating:** 2

**Summary Of Contributions:**

This paper applied differentiable architecture search to inverse problems. The experiments showed problems of one-shot neural architecture search, including a large variance, the performance gap between supernet and final performance, and the importance of hyperparameters. The interesting point of this paper is the design of search spaces. Harmful operations are included to check if the search algorithm can avoid harmful operations.

**Actions Required To Increase Overall Recommendation:**

I do not have room to adjust my score. However, I am still happy to hear back from the authors and will reconsider it.

**Overall Review:**

[+] Add harmful operations to evaluate search algorithms.

[-] I did not find new experimental results. Experiments are not complete and solid.

**Potential Impact On The Field Of Automl:**

I do not expect a lot of discussions of this paper. This paper raised some problems of one-shot architecture search. These problems are not new and well-know to the AutoML community. It is common sense that there is a gap between the supernet and sub-nets. It is well known that HPO is very important and sometimes HPO brings more performance gain than NAS. The authors re-discovered these problems of one-shot NAS on inverse problem tasks.

**Reproducibility (Optional):**

I feel it is easy to reproduce it.

**Review Confidence:**

4: You are confident in your assessment, but not absolutely certain. It is unlikely, but not impossible, that you did not understand some parts of the submission or that you are unfamiliar with some pieces of related work.

**Review Rating:**

3: Reject: For instance, a paper with technical flaws, weak impact, and/or weak evaluation.

**Review Summary:**

This paper re-discovered problems of one-shot search. It is not necessary to use inverse problems to repeat these problems again. I hope the authors can solve these problems when they apply differentiable search to inverse problems.

---

> ### Author Response · Authors · 2023-05-02
> **Response to Reviewer**
>
> Thank you for your feedback. To the best of our knowledge, existing work towards analyzing "one-shot search" using DAS has so far focused on well studied image classification problems, leading to the following limitations: (1) each search run is expensive so that the strive for significant (negative) results is heavily challenged, and (2) the search hyperparameters are already established so that an unbiased analysis can not be provided.
>
> Our paper closes this gap.
>
> Are there particular studies that you feel are not well represented in our discussion?
> We would have been happy to clarify and include these.
> Further, we would happily respond to additional actionable suggestions.
>
> Thank you for your time and consideration.

---

### Review · Reproducibility_Reviewer_ac9C · 2023-04-13

**Completeness Of Code And Dataset Supplement Rating:** 4
**Usability And Ease Of Reproducibility Rating:** 4

**Actions Required To Increase The Reproducibility And Overall Recommendation:**

A suggestion is to provide more details in read me file about different shell files to run. I saw some details in the shell files that I believe are better suited to be put in read me file.

**Completeness Of Code And Dataset Supplement:**

The code seems complete to me and I was able to run it. I believe the results of the paper are reproducible with provided code.

**Overall Reproducibility Review:**

The code seems well written and is executable without a problem. There is no major issues with code, but can benefit from a more detailed instruction.

**Review Confidence:**

4: You are confident in your assessment, but not absolutely certain. It is unlikely, but not impossible, that you did not understand some parts of the submission or that you are unfamiliar with some pieces of the code or data.

**Review Rating:**

10: Exceptional, this paper is reproducible at the push of a button and a model for the community.

**Review Summary:**

The code provided contains all the major experiments of the paper and it is easily executable. I was not able to reproduce the results given in the paper however I believe the code is sufficient to achieve this.

**Summary Of Necessary Code And Dataset Supplement:**

 Reimplementation of DARTS (Liu et al., 2019) for Differentiable Architecture Search (DAS)  is used for inverse problems in this paper. Search space is designed  for sequential non-sequential data and data generation is performed for experiments by random sampling from Gaussian distribution (with adding Gaussian noise, blurring, and downsampling). DAS is performed for 75 trials as well as several baselines, and the correlation (peak signal to noise ratio (PSNR)) for one-shot search vs. architecture performance) are plotted and a regression if performed on this data. BOHB (Falkner et al., 2018) is applied for hyperparameter optimization and compared with several baselines.

**Usability And Ease Of Reproducibility:**

I was able to run the provided code. I did not reproduce the same results, however I believe that it can be reproduced by the given code. I think the provided code is very good, however it can be improved by more detailed documentation.

---

> ### Author Response · Authors · 2023-05-02
> **Response to Reproducibility Reviewer**
>
> We thank the reviewer for the feedback of providing more details in the README file. We updated our README  file and added the script for the new experiments on image data (Sec. 4.4). The new code is provided with the revised version.
>
> We thank the reviewer again for the input and hope we addressed all remaining concerns.

---

> > ### Comment · Reproducibility_Reviewer_ac9C · 2023-05-08
> > **Respons3 to Paper25 Authors**
> >
> > Thank you for the revisions of the code as well as the code for new experiments. I reconfirm my rating for the reproducibility as exceptional.

---

### Official Review · Reviewer_GfZK · 2023-04-13

**Potential Impact On The Field Of Automl Rating:** 2
**Technical Quality And Correctness Rating:** 2
**Clarity Rating:** 3

**Summary Of Contributions:**

The paper takes a closer look at Differentiable Architecture Search and analyses its performance in the context of inverse problems: more specifically using cosine waves of varying magnitude, amplitude and offset, with the goal of searching for models that can recover the samples from distorted measurements. The key findings as described in the paper are 1) DAS has large variance for inverse problems, 2) selection of hyperparameters has a large influence on the performance of DAS, 3) the performance of the weight-sharing architecture used during training does not reflect the final performance of the found architecture well.


**Actions Required To Increase Overall Recommendation:**

Key areas that could improve the paper:
* Update how the results are reported in the tables - especially include stds or confidence intervals
* Do some experiments that would give us insights how the findings transfer from DAS for simple inverse problems to more complex and realistic problem settings


**Clarity:**

The paper is clear and easy to understand. There are several aspects that could be improved to improve the clarity further:

* Provide more details about how the validation set is constructed.
* Visualization of the search space is skewed (Figure 1 and 2)
* Proof-read the supplementary for typos
* Table 1 typo - there is sec for runtime, but the format is min:sec - would also be good to add to Table 2
* How were the manually designed HPs selected?
* Figure 1 could ideally be closer to where it is mentioned


**Overall Review:**

Positives:
* The general idea of the paper is interesting: take a deeper look at the performance of differentiable NAS, using lightweight inverse problems setting.
* The paper identifies specific points that highlight potential limitations of DAS: instability across trials, instability across HPO tuning of the NAS hyperparameters, differences between one-shot performance ranking and full re-training.
* The paper comes with a new problem setting that could be potentially of interest to other NAS researchers thanks to being lightweight: inverse problems.
* The paper is well-written and easy to read.

Negatives:
* The significance of the main findings - instability across trials and instability across sets of hyperparameters - may be overrated given the results presented in the paper. Disparity between one-shot performance and re-training performance appears to be investigated already in multiple studies based on Section 2: Related Work, Stability of DAS.
* Analysis of the results may not be the most appropriate to draw the given conclusions: the paper reports max, mean and median across 75 or 100 repetitions, but reporting stds or confidence intervals may be more appropriate to judge if the main messages of the paper hold and to what extent. Min instead of max is likely to be more interesting as min corresponds to worst-case performance here.
* Given that the general goal seems to be to find general findings about differentiable NAS methods rather than just within the context of inverse problems, some evaluation also on other problems (e.g. CIFAR-10 classification) would be helpful. Are there any parts of the analysis that would be viable also on other problems than inverse problems so that we can find how general the findings can be? Generally it would be good to provide some intuitions for how well the findings from the inverse problem transfer to other settings of interest.
* All analysis seems to be only done on validation data - test data (especially in combination with re-training) would appear more appropriate.
* When not using harmful operations in the search space, DAS is worse than pure random sampling within the sequential search space.
* When using all operations in Table 2, there is only DAS and random sampling. Is it really the case there is no other simple baseline that could be evaluated here too?

Other questions:
* To what extent is including harmful operations in the search space realistic? It certainly makes sense as part of evaluation (the method should avoid those), but if there is not much hope the operations would help, then these normally would not be included in well-designed search space? If that is the case then a good ops search scenario is more realistic?
* What are the times to run the experiments? Information is provided in Table 1, but it is missing in Table 2. Also times to run HPO for DAS would be useful to know.
* Why are there 75 trials in Table 1 and 100 in Table 2?


**Potential Impact On The Field Of Automl:**

The key message of the paper is that differentiable architecture search may be unstable in terms of 1) repeating the experiments multiple times, 2) selection of hyperparameters, 3) performance after retraining. While these may be important to know for researchers working with such methods, I believe that further discussion of the results provided in the paper is needed as points 1 and 2 may not be sufficiently supported with the results. Instability after re-training (point 3) has been mentioned in the literature already in multiple studies (see Stability of DAS of 2 Related work).


**Reproducibility (Optional):**

The code is provided, including commands to run the experiments, so the experiments should be reproducible. The fact that the experiments are lightweight is also helpful in this.


**Review Confidence:**

4: You are confident in your assessment, but not absolutely certain. It is unlikely, but not impossible, that you did not understand some parts of the submission or that you are unfamiliar with some pieces of related work.

**Review Rating:**

5: Borderline Leaning Reject: Technically sound paper where reasons to reject nonetheless outweigh reasons to accept. Please use sparingly.

**Review Summary:**

The paper raises interesting points about differentiable NAS methods, but there are questions about how these findings follow from the results provided. I’m happy to increase the rating and recommend acceptance if the discussion resolves my concerns.


**Technical Quality And Correctness:**

I have some questions about the interpretation of the results and how they lead to the findings.

For repetition of the experiments multiple times: max, mean and median is reported across 75 or 100 trials in various tables. To be able to compare the stability across repetitions, reporting standard deviation or confidence interval may be more appropriate - is it large compared to the magnitude of the mean? In some cases the maximum is indeed significantly larger than the mean or median, but in many cases it is relatively similar, so I am not completely sure about the claim about large variance. Also reporting min value may be more interesting than max because we are trying to maximize the performance and worst-case scenarios can be seen as more interesting.

Regarding HPO stability: this again suffers from not analysing standard deviations or confidence intervals. The instability generally happens when applying the hyperparameters out of the domain: e.g. tuning it for blur and deploying for downsampling. When applying the right hyperparameters the results appear to be quite stable, so I am not sure about the significance of saying that selection of hyperparameters has a large influence on the performance of DAS. Some level of impact is expected by default too.

---

> ### Author Response · Authors · 2023-05-02
> **Response to Reviewer I**
>
>
> We appreciate the reviewer’s acknowledgment of the potential of our work and the suggestions made.
>
>
> **1. For repetition of the experiments multiple times: [...] To be able to compare the stability across repetitions, reporting standard deviation or confidence interval may be more appropriate. Analysis of the results may not be the most appropriate to draw the given conclusions)**
>
> We added minimal PSNR values as well as the standard deviation to all tables in our paper. Indeed, we can now further analyze the stability across all trials with regard to worst-case scenarios. For the sequential search space, we see that the random baseline has the highest minimal value, whereas this picture changes for the non-sequential search space, in which DAS has the highest minimal value.
>
>
>  **2. When applying the right hyperparameters the results appear to be quite stable, so I am not sure about the significance of saying that the selection of hyperparameters has a large influence on the performance of DAS.**
>
> Also here, we can analyze the stability of DAS with regard to worst-case scenarios in a better way, after including the minimal PSNR and the standard deviations. As we can see in Tab. 3, the standard deviation for the blur data formation is rather small for all hyperparameter sets and operation sets, whereas the minimal value for the supposedly best hyperparameter set (BOHB-Blur) is the lowest but has the highest standard deviation. Therefore, different hyperparameter sets do have a large influence on the stability of the DAS search approach.
>
>  **3.Provide more details about how the validation set is constructed. All analysis seems to be only done on validation data - test data (especially in combination with re-training) would appear more appropriate.**
>
>  All models are tested on random 1D signals contaminated by noise that are generated on-the-fly for every validation. More details can be found in Section 4.1. Because of this sampling procedure there is no re-use of validation data and all analysis could be equally understood as being run on test data.
>
>  **4. Visualization of the search space is skewed (Figure 1 and 2)**
>
> We updated Figures 1 and 2.
>
>  **5. Proof-read the supplementary for typos**
>
> We have proof-read the supplementary again.
>
>  **6. Table 1 typo - there is sec for runtime, but the format is min:sec - would also be good to add to Table 2**
>
>  We changed the runtime format, to make it more clear and also added the runtimes to all tables.
>
> **7. How were the manually designed HPs selected?**
> Manually designed hyperparameters were chosen based on best practices in this domain. Here, they correspond broadly to the hyperparameters for DnCNN, for training-sepcific hyperparameters. DAS hyperparameters are chosen based on the original implementation.
>
> **8. Figure 1 could ideally be closer to where it is mentioned**
>
> We updated that.
>
>
> **9. Instability across trials and instability across sets of hyperparameters - may be overrated, already  other related work investigated this.**
>
> We agree, that previous work already pointed out the rank disorder and poor test generalization using DAS for image classification task. In this paper, we focus on analyzing, if these problems also hold in other domains, i.e., inverse problems, and if changes like search space, hyperparameters, and also improvements on the random initialization (Appendix B in the revised version) improve the named problems. We can indeed see, that (i) a proper search space for the task of interest, in our case the sequential search space, influences the performance outcome substantially, and (ii) overcoming the random initialization and just searching for the operations at each layer by using a single-level optimization further improves the test generalization.
>
> **10. Given that the general goal seems to be to find general findings about differentiable NAS methods rather than just within the context of inverse problems, some evaluation also on other problems would be helpful. Are there any parts of the analysis that would be viable also on other problems than inverse problems so that we can find how general the findings can be?**
>
> In general, this study shows, that the problems which are already known with DAS on image classification also transfer to this domain of inverse problems. In this paper, we analyze DAS in the domain of inverse problems on a more fundamental level and investigate if the same problem also occur here. We show, that also in this simple 1D example, we face similar problems as the domain of image classification, which is the most commonly used domain for DAS: the ranking disorder and the poor test generalization.

---

> > ### Author Response · Authors · 2023-05-02
> > **Response to Reviewer II**
> >
> >
> > **11.  Is it really the case there is no other simple baseline that could be evaluated here too?**
> >
> > In this paper, the focus is on DAS and its ability to perform one search run to find a well-performing architecture in the domain of inverse problems. In order to evaluate its ability for being a one-shot model, we focused on the most simple existing baseline. Since DAS fails in some cases to improve already over this simple baseline, we did not include other baselines.
> >
> > **12. To what extent is including harmful operations in the search space realistic?**
> >
> > We consider this point relevant for several reasons, e.g.: (1) One fundamental hope is that NAS would facilitate the design of well performing neural architectures even for non ML experts, e.g.~companies in need for a solution to a particular practical problem. In a new scenario, could a non-expert confidently exclude harmful operations? (2) In terms of robustness, we would argue that a NAS approach has to handle this case of harmful operations in a predictable way. When applied to a new search space, small programming mistakes can easily turn a good operation into a poor one that should not be chosen. (3) Would it not be disappointing, from a theoretical perspective, if a methods towards optimizing a neural architecture fails to identify poor operations? To our understanding, this should be an easy task for a practically relevant tool.
> >
> >
> > **13. What are the times to run the experiments? Information is provided in Table 1, but it is missing in Table 2.
> > Why are there 75 trials in Table 1 and 100 in Table 2?**
> >
> > We updated Tab. 2 with the run times. Given the overall worse performance on the non-sequential search space, we conducted more runs in this search space to better analyze the general convergence behavior.
> >
> > **14. Do some experiments that would give us insights how the findings transfer from DAS for simple inverse problems to more complex and realistic problem settings.**
> >
> > We additionally evaluated DAS on the Berkeley Segmentation Dataset and Benchmark (BSDS300) for image denoising in Sec. 4.4 in our updated paper. We compare DAS to the DnCNN architecture (Zhang et al., 2017) and to a random sampling baseline, where the latter random sampling approach improves over both, DAS and DnCNN. However, we want to point out here, that this study is possible since the search and evaluation times are rather small and thus allow for these exhaustive evaluations, including hyperparameter search. This study is not easily feasible for larger datasets.
> >
> >
> > We thank the reviewer again for the input and hope we addressed all remaining concerns.

---

> > > ### Comment · Reviewer_GfZK · 2023-05-08
> > > **Response**
> > >
> > > Thanks for the additional details, explanations and experiments, it certainly improves the quality of the paper. However, I still have some key worries about the paper, especially after the stdevs have been added.
> > > * Given the magnitude of the stdevs, it is quite hard to say if we can make some of the claims in the paper - e.g. “DAS improves over a random search baseline by a significant margin”.
> > > * Regarding the size of variance and instability: I’d agree the variance is large when including all operations, but not when including only the good ones. After adding stdevs, various sets of hyperparameters seem to give similar results, contrary to saying “different hyperparameter sets do have a large influence on the stability”.
> > > * Simple baselines: if random search is too expensive, you could run something simple like selecting the best model after running one (or more generally N) training iteration(s) with each candidate configuration. Maybe this would be competitive in the scenario where random search is not evaluated due to too large time costs.
> > > * There is also the broader issue that the paper seems to mainly identify NAS challenges already known in literature.
> > >
> > > For now I maintain my rating, primarily because I do not find the claims in the paper sufficiently supported with the results.

---

> > > > ### Author Response · Authors · 2023-05-09
> > > > **Response to Reviewer**
> > > >
> > > > Dear Reviewer, thank you for the question.
> > > > The observation of large standard deviations in DAS is actually one of the strongest motivations of our study setting. As specified for example in lines 51-53 in our paper, the variance in the performance of DAS over different runs calls for an evaluation in a statistically significant way. To do so, we propose the computationally manageable signal reconstruction setting and report results over a sufficiently large amount of results. This is not reflected in the standard deviation but in the standard error (std err), which also takes the number of runs into account.
> > > >
> > > > For example, consider the std err of the results from Table 1 (e.g. blur, only good ops) where we evaluate 75 runs, and let us compare DAS against Random Sampling as you suggest. For DAS, the std dev is *1.03* so the std err is *~0.12*. Thus, statistically, the result of DAS will be within the interval *[21.33, 21.79] with a confidence of 95%*. Random Sampling with std dev of *0.77* has a std error of *~0.09*. The result of Random Sampling is thus expected to be within the interval *[21.96, 22.14]  with confidence of 95%*.
> > > > In fact, we think our paper will be useful to the community in highlighting exactly this need for a large number of runs, in order to prove the statistical significance of results of DAS.
> > > >
> > > > If it is helpful, we can of course include the standard error instead of the standard deviation in all our tables and and the explicit 95% confidence intervals of our evaluations in the appendix.
> > > >
> > > > Does this address your concern? Are there any further questions we could address?

---

> > > > > ### Comment · Reviewer_GfZK · 2023-05-09
> > > > > **Response**
> > > > >
> > > > > Thanks, statistical evaluation using std err would be more appropriate here and would support the claims. For simplicity it could be enough to only report mean and std error, but of course the other metrics can be included too (especially the minimum). However, even after using more appropriate statistical evaluation the other concerns remain (baselines, findings mentioned here already known in literature, ...).

---

### Author Response · Authors · 2023-05-02
**General Response**

We thank the reviewers for their feedback and their acknowledgment, that this paper provides an *interesting idea* (R GfZK) with a *good study* (R wSGU) on a *fundamental level* (R 4Z5E) for DAS in a new domain of inverse problems, being a *very important topic* (R 4Z5E).

We agree, that additional information about the results, such as minimal value and standard deviation, is helpful for further investigation of the stability of DAS and the random sampling-based baselines. Therefore, we included these values in each table.

In addition, we provided experimental results on image data (BSDS for image reconstruction) in which we compare DAS to random sampling and the common network for inverse problems, DnCNN, in Sec. 4.4.

We thank the reviewers for their input and hope we addressed all remaining concerns.